# Fibroblast mechanotransduction network predicts targets for mechano-adaptive infarct therapies

Jesse D Rogers, William J Richardson*

Department of Bioengineering; Clemson University, Clemson, United States

**Abstract** Regional control of fibrosis after myocardial infarction is critical for maintaining structural integrity in the infarct while preventing collagen accumulation in non-infarcted areas. Cardiac fibroblasts modulate matrix turnover in response to biochemical and biomechanical cues, but the complex interactions between signaling pathways confound efforts to develop therapies for regional scar formation. We employed a logic-based ordinary differential equation model of fibroblast mechano-chemo signal transduction to predict matrix protein expression in response to canonical biochemical stimuli and mechanical tension. Functional analysis of mechano-chemo interactions showed extensive pathway crosstalk with tension amplifying, dampening, or reversing responses to biochemical stimuli. Comprehensive drug target screens identified 13 mechano-adaptive therapies that promote matrix accumulation in regions where it is needed and reduce matrix levels in regions where it is not needed. Our predictions suggest that mechano-chemo interactions likely mediate cell behavior across many tissues and demonstrate the utility of multi-pathway signaling networks in discovering therapies for context-specific disease states.

## Editor's evaluation

This paper presents a computational network model of fibroblast signalling in order to identify drug combinations that might be useful targets for controlling cardiac fibrosis, with application to treating myocardial infarction. The approach clearly has merit and is a potentially powerful way to make progress in understanding effects of interventions in situations where regional variations in tension and biochemical alterations have to date frustrated many attempts of understanding and rational treatment.

*For correspondence:
wricha4@clemson.edu

Competing interest: The authors declare that no competing interests exist.

## Introduction

Controlling cardiac fibrosis remains a major challenge in developing long-term treatments for patients suffering from myocardial infarction (MI). For a proportion of the over 800,000 patients diagnosed per year (*Benjamin, 2019*), excessive hypertrophy and scar formation in the non-infarcted myocardium can result in the development of heart failure, with decreases in contractility, myocardial conductivity, and pump function leading to poor rates of survival (*Shah et al., 2017*). Currently prescribed therapeutics for MI patients are designed to reduce hypertrophy via several neurohormonal mechanisms including inhibition of angiotensin II (AngII) and norepinephrine (NE) or increasing the bioavailability of natriuretic peptides (NPs) (*Ponikowski et al., 2016*), but no drugs have been approved for treating cardiac fibrosis directly. Notably, scar formation serves a reparative function in post-MI wound healing by replacing necrotic myocardium and preventing infarct expansion and cardiac rupture (*Weber et al., 2013*). Therefore, developing therapeutics that limit excessive fibrosis to prevent the development of

heart failure while preserving beneficial scar tissue at the infarct site would be a critical step toward improving long-term patient outcomes (*Richardson et al., 2021*).

Tissue remodeling in the myocardium is a dynamic process in which the accumulation of extracellular matrix proteins (e.g. collagens I and III, fibronectin) and matricellular proteins are balanced by degradation via proteases such as matrix metalloproteinases (MMPs). This balance is mediated largely by cardiac fibroblasts, which infiltrate the infarct site and assume an activated, synthetic phenotype to promote matrix accumulation (*Chen and Frangogiannis, 2013*). A wide variety of biochemical factors regulate fibroblast behavior such as AngII, transforming growth factor-β (TGFβ1), endothelin-1 (ET1), and various inflammatory cytokines (*Leask, 2010*; *Turner, 2014*), but a growing body of research has demonstrated that biomechanical factors such as circumferential stretch and tissue stiffness are also critical determinants of fibroblast behavior (*Saucerman et al., 2019*). Parallel transduction of biochemical and biomechanical stimuli is integrated by fibroblasts through a complex signaling network in which pathways interact via systems-level crosstalk and feedback mechanisms (*Hinz, 2015*; *Saraswati et al., 2020*), making prediction and control of fibroblast behavior in the post-MI environment highly challenging.

A variety of computational approaches have been used to investigate molecular networks including mechanistic models, statistical models, and artificial intelligence approaches. Ordinary differential equation (ODE) network models offer the ability to provide a mechanistic basis for disease progression while accounting for complex system dynamics. One advantage of mechanistic networks over 'black-box' statistical and machine learning models is that they directly facilitate the discovery of influential pathways and druggable targets. Several recent studies have employed a normalized Hill-ODE approach for network modeling, which offers a balance between fully mechanistic network models (i.e. mass action kinetics that require challenging measurements of many rate parameters) and overly simplified Boolean models (which can have difficulty discerning the nonlinear interactions between competing cues with different quantitative contributions) (*Irons and Humphrey, 2020*; *Kraeutler et al., 2010*; *Ryall et al., 2012*; *Tan et al., 2017*; *Zeigler et al., 2016*). Using a system of ODEs to approximate changes in activation of signaling proteins, cellular responses to both individual stimuli and combinations of stimuli can be predicted, providing a basis for targeted experimental validation of new mechanisms-of-action or treatments with therapeutic potential.

In this current study, we expanded a previously published model of fibroblast mechano-chemo signal transduction capable of accurately predicting fibrosis-related protein expression in response to canonical biochemical factors and mechanical tension (*Zeigler et al., 2016*). We examined the extent and dynamics of mechano-chemo interactions by simulating biochemical dose-response behavior under varying levels of mechanical stimulation, finding that tension amplified, dampened, or reversed fibroblast responses to individual biochemical cues. Comprehensive simulations of fibroblast responses to over 23,000 combinatory drug targets in low- or high-tension contexts identified several drug combinations that adapted fibrotic activity to the local mechanical state, demonstrating the potential of this model to act as a screening tool for targeted therapeutics.

## Results

### Development of fibroblast mechanotransduction network

To develop a fibroblast signaling model capable of sensing and transducing dynamic tension, we performed a manual literature search of intracellular signaling reactions related to fibroblast mechanotransduction and integrated our search with an existing model of fibroblast signaling (*Zeigler et al., 2016*). Our extended model includes nine cytokine inputs known to mediate fibroblast activation post-MI as well as mechanical tension, which is transduced via multiple mechanosensors including integrins, stretch-activated ion channels, and the G-coupled protein receptor angiotensin type I receptor (AT1R) (*Figure 1*, and see *Figure 1—figure supplement 1* for nodes added to the previous network). We also added several outputs that have been recently shown to mediate myocardial remodeling, including proMMPs 3, 8, and 12, tenascin-c (TNC), osteopontin, and thrombospondin-4 (*Frolova et al., 2012*; *Iyer et al., 2015*; *Singh et al., 2010*; *Song et al., 2017*), as well as autocrine feedback mechanisms found to regulate fibroblast activation (*Ashizawa et al., 1996*; *Midwood and Schwarzbauer, 2002*; *Zent and Guo, 2018*). The total network, consisting of 109 nodes and 174 reactions, was implemented as a logic-based ODE model using the open-source Netflux package as

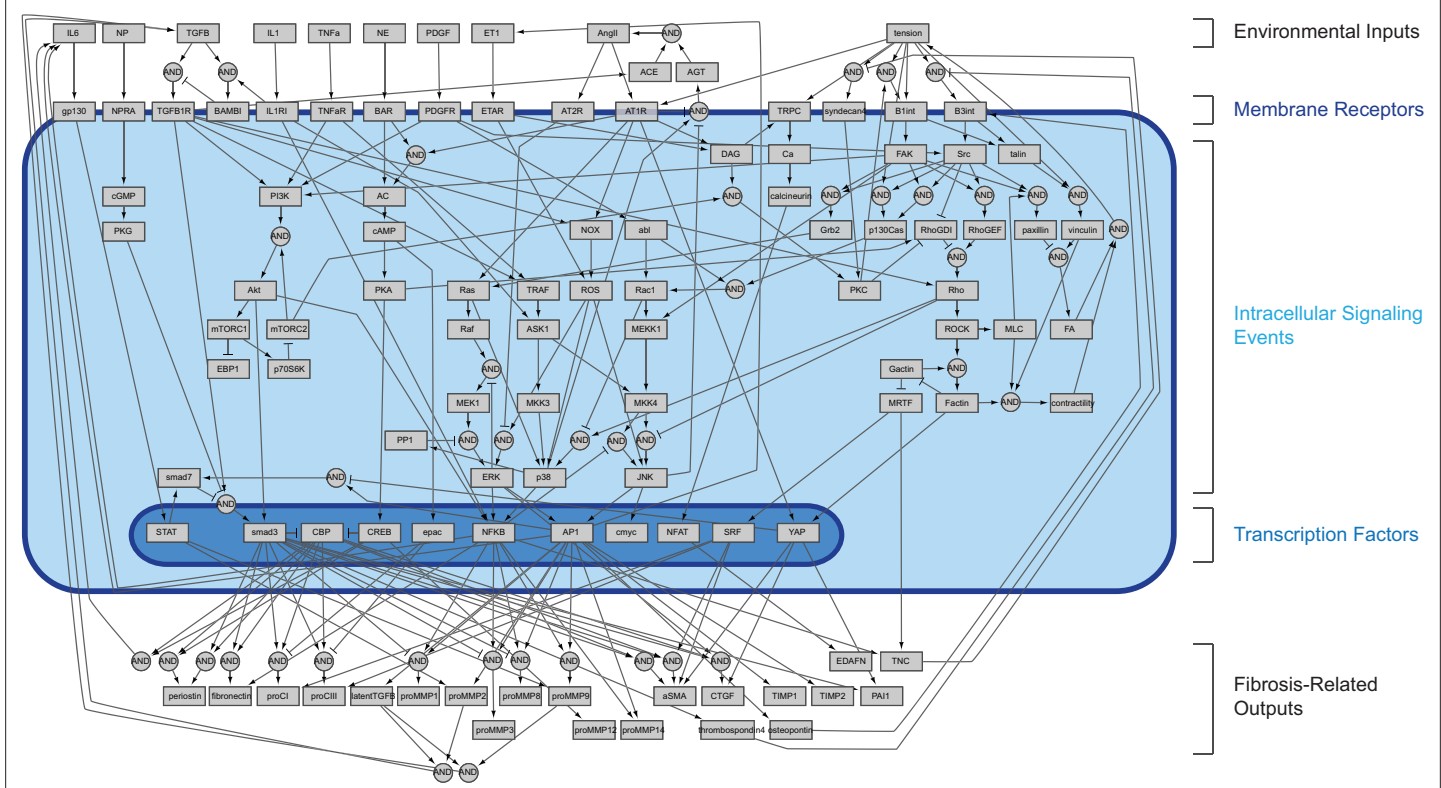

**Figure 1.** Schematic of expanded logic-based ordinary differential equation model of cardiac fibroblast chemo/mechanotransduction. Extracellular stimuli, intracellular signaling species, transcription factors, and fibroblast-secreted outputs are represented as nodes (boxes, 109 total). Directed edges represent activating and inhibiting reactions (arrows and T junctions respectively, 174 total) with AND logic indicated by circular nodes.

The online version of this article includes the following figure supplement(s) for figure 1:

**Figure supplement 1.** Schematic of fibroblast chemo-mechanotransduction network modifications for current study.

previously described (*Kraeutler et al., 2010*) in which node activity levels and reaction weights are normalized values between 0 and 1 (e.g. an activity of 0.5 represents 50% of maximum activity). All model species, reaction logic, default reaction parameters, and supporting references can be found in *Supplementary file 1*.

Model predictions were validated against an independent set of experimental studies by comparing qualitative changes (i.e. increase, decrease, or no change) in node activation to changes in protein activity measured in vitro (see Materials and methods, Model validation section for full description). A curated body of experimental literature kept separate from model construction was used for validation, and most studies were additionally kept separate from each other in terms of authorship, with only 4 of 47 studies containing repeat authors from a single group and 2 of 47 studies containing a repeat author from a separate group, accounting for 10.2% and 1.7% of the total validation set for each group. Parameter sweeps for half-maximal effective concentrations (EC50) and Hill coefficients (n) identified default reaction parameters for optimizing predictive accuracy (*Figure 2—figure supplement 1*). For fibroblast-secreted outputs, simulated responses to biochemical and mechanical stimuli correctly predicted qualitative changes in protein secretion in 81.8% (63/77) of simulations (*Figure 2A*). Additionally, the model correctly predicted qualitative changes in activation levels of canonical signaling intermediates for 80.5% (33/41) of simulations compared to experimental studies (*Figure 2B*). Consistent with previous studies, the model-predicted increased expression of collagens, matricellular proteins, and protease inhibitors with exposure to canonical agonists TGFβ1, AngII, platelet-derived growth factor (PDGF), tumor necrosis factor-α (TNFα), and mechanical tension, and secretion of these pro-fibrotic species was accompanied by increased MMP production. Disagreement between model predictions and experimental studies were typically found for studies that reported either decreases or no change in output expression with interleukin-1β (IL1), interleukin-6 (IL6), or NE

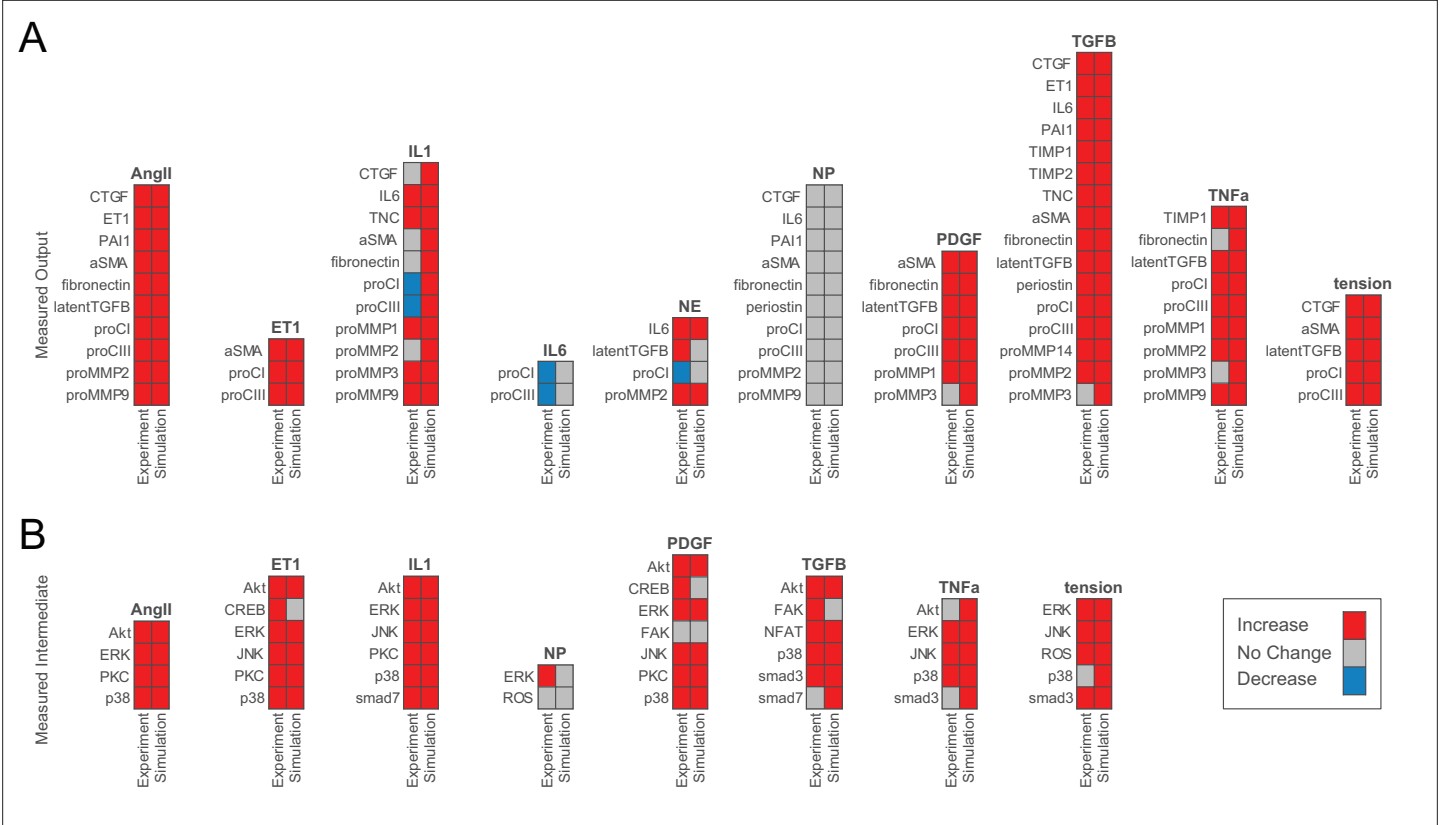

**Figure 2.** Model accurately predicts qualitative changes in output expression and intermediate activity as measured by independent in vitro studies. (**A**) Expression of cell-secreted outputs and phenotypic markers (i.e. αSMA) were predicted in response to single biochemical stimuli and mechanical tension and compared to an independent set of experimental studies found in literature (47 papers total). (**B**) Activation of intracellular intermediates representing major signaling pathways was predicted in response to single stimuli and compared to independent experimental studies measuring changes in activity (e.g. by phosphorylation, 27 papers total). All model predictions were categorized based on a ±5% change in activity levels compared to baseline conditions.

The online version of this article includes the following figure supplement(s) for figure 2:

**Figure supplement 1.** Optimization of reaction parameters for maximizing accuracy of qualitative predictions.

simulation. However, as described in the next section, these incorrect model predictions could be rectified with the experimentally observed decreases under other input contexts.

We additionally investigated the sensitivity of predictive accuracy against the threshold change in activity levels used to classify model predictions by repeating the above analysis for thresholds between 1% and 10% change in activity. We found that the accuracies of both output and intermediate node predictions were robust to the threshold change in activity, as output accuracy deviated by ±1.3% from the median at the minimum and maximum thresholds, and intermediate accuracy did not change across all simulations (*Figure 2—figure supplement 1C*). Output and intermediate accuracies remained above 80% across all threshold levels, and the same analysis performed an existing model of fibroblast signaling (*Zeigler et al., 2016*) showed consistent increases in accuracy across the thresholds tested, demonstrating that the current model improves on predictive accuracy across a range of activity thresholds.

## Mechanical tension amplifies, dampens, or reverses network responses to biochemical stimuli

Several previous experimental studies have established that combined biochemical and biomechanical signals can interact to produce context-dependent responses from fibroblasts (*Bishop et al., 1998*; *Kural and Billiar, 2016*; *Lindahl et al., 2002*; *Merryman et al., 2007*; *Speight et al., 2016*). But the full scope and mechanisms behind these interactions are mostly unclear. We investigated interactions

between tension and individual biochemical stimuli computationally by simulating dose-response relationships between biochemical inputs and all model nodes under increasing levels of tension, using area under the curve measurements (AUCs) integrated over all biochemical input doses as a metric for total change in activity for each level of tension (see Materials and methods, Mechano-chemo interaction analysis section for full description). We characterized interactions based on changes in each AUC from basal tension to increased levels of tension, categorizing nodes with increased AUC levels with tension as being *amplified*, nodes with decreased AUC levels as being *dampened*, and nodes that changed the direction of activation with tension (i.e. from net activation to net inhibition or vice versa) as being *reversed*.

We found that these interactions were dependent on both the individual inputs as well as the level of tension. Of the nine biochemical model inputs, AngII, TGFβ1, IL1, TNFα, PDGF, and ET1 all demonstrated sizable changes in dose-response behavior with various levels of tension, based on a 5% change in AUC levels. Using this threshold, a tension level of 0.5 (i.e. 50% of maximum stimulation) amplified, dampened, or reversed an average 56.7% of node AUCs toward these six inputs compared to basal levels of tension (*Figure 3A*), and a tension level of 0.9 (i.e. 90% of maximum stimulation) sizably altered an average 60.4% of node AUCs (*Figure 3B*). We additionally compared the distribution of changes in AUCs across all nodes between selected levels of tension using two-sample Kolmogorov-Smirnov tests (see Materials and methods, Mechano-chemo interaction analysis for full description), finding that tension significantly altered distributions of dose-response behavior of these six inputs at tension levels of both 0.5 and 0.9 (*Figure 3—figure supplement 1* and *Table 1*). Conversely, tension altered the dose-response behavior of very few nodes with IL6, NE, and NP stimuli, with only 7.03% of node AUCs on average changing sizably toward these inputs for a tension level of 0.9 (*Figure 3—figure supplement 1*). We found that although tension levels of 0.5 and 0.9 significantly altered probability distributions of changes in AUC for these inputs (*Table 1*), the relatively small population of nodes with sizable changes suggests that these inputs were largely unchanged with tension. In support of their insensitivity to tension, we found that IL6, NE, and NP all displayed lower degrees of network connectivity compared to the remaining inputs as evidenced by lower measures of closeness centrality and higher measures of average shortest path length and eccentricity (*Figure 3—figure supplement 1*). While our previous work has shown that these measures do not correlate with functional measures such as sensitivity studies (*Zeigler et al., 2016*), these topological trends point to a less-influential role of tension for inputs that are relatively less connected to mechanotransduction pathways in the network.

Within inputs that demonstrated tension-modulated dose-response behavior, the level of tension applied mediated different types of interactions. Moderate tension levels (reaction weights 0.2–0.5) largely amplified the effects of inputs; of the six inputs demonstrating tension-mediated changes in dose-response behavior, 51.8% of node AUCs were amplified at a tension level of 0.5 on average while 4.43% and 0.46% of node AUCs were dampened or reversed on average, respectively (*Figure 3A*). High levels of tension (reaction weights 0.6–0.9) largely dampened network responses to these inputs, with 56.4% of nodes on average decreasing AUCs with the same six inputs compared to 2.75% and 1.22% of node AUCs showing sizable amplification or reversal, respectively (*Figure 3B*). Repeated analyses with incremental levels of tension suggested that this change in tension-mediated behavior occurred at a tension reaction weight of 0.6, with an average 40.3 ± 12.1 nodes changing from amplification to dampening at this level for these inputs compared to a reaction weight of 0.5 (*Figure 3—figure supplement 1*). These changes in sensitivity were indicative of network interactions between tension and biochemical stimuli; we observed that NE-mediated inhibition of procollagen I expression was amplified by tension (*Figure 3C*), which can be explained by the logical requirement for tension-induced smad3 activation for effective inhibition by cAMP response element-binding protein (CREB) (*Figure 3D*). We also found that tension-mediated desensitization of procollagen I expression toward TGFβ1 can be explained mechanistically by a common autocrine feedback loop integrating both tension and TGFβ1 signals via a reactive oxygen species (ROS)-activator protein 1 (AP1) signaling axis (*Figure 3E–F*).

Notably, 4.40% of nodes reversed the direction of dose-response curves with IL1 stimulation with a tension reaction weight of 0.6 or greater compared to baseline tension levels, including intermediate nodes involved in TGFβ1 signaling (TGFβ1R, NOX, ROS), mitogen-activated protein kinase (MAPK) signaling (TRAF, ASK1, MKK3, MKK4, ERK), as well as the expression of procollagen I (*Figure 3G*).

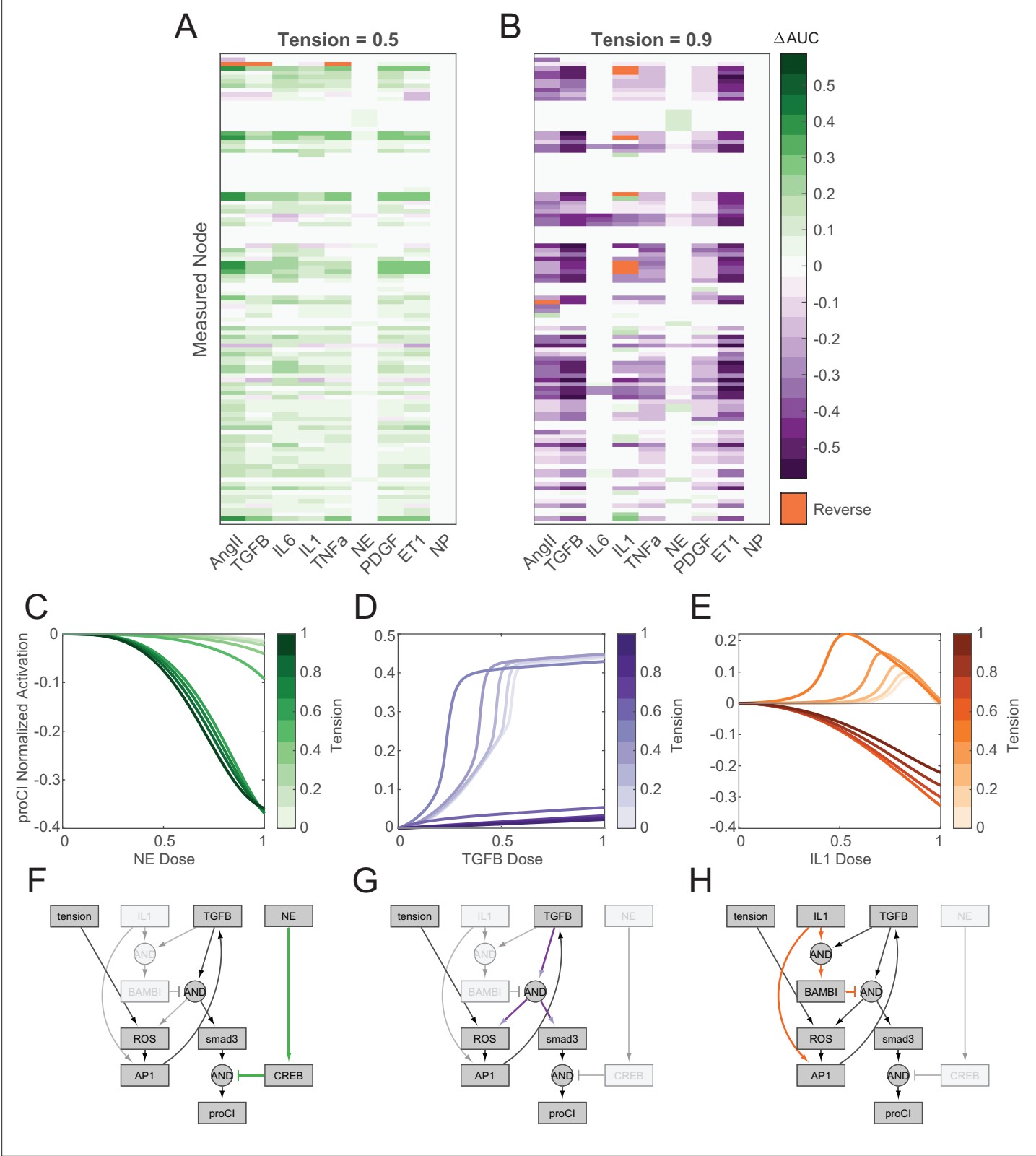

**Figure 3.** Network-wide responses to biochemical stimuli are dependent on the tensional context. (**A–B**) Dose-response curves were simulated under elevated levels of tension with incremental doses of each biochemical input, and changes in area under normalized dose-response curves (ΔAUC) were calculated for all nodes in the model in comparison to basal dose-response simulations (i.e. tension = 0.1). Reversal cases were identified based on opposing AUC signs between basal and elevated tension simulations, that is, nodes that exhibited a positive AUC with basal tension and a negative

*Figure 3 continued on next page*

*Figure 3 continued*

AUC with elevated tension, and vice versa. (**C–E**) Examples of mechano-chemo interaction categories identified from network-wide analysis in (**A**) and (**B**). Normalized dose-response curves for procollagen I (proCI) expression are shown following simulations with incremental doses of norepinephrine (NE) (**A**), transforming growth factor-β (TGFβ1) (**B**), and interleukin-1β (IL1) (**C**) under static levels of tension. (**F–H**) Sub-networks regulating the mechano-chemo interactions shown in (**C–E**) were determined by manual examination of node activity levels following dose-response simulations using NE (**F**), TGFβ1 (**G**), and IL1 (**H**) stimuli under basal and elevated levels of tension.

The online version of this article includes the following figure supplement(s) for figure 3:

**Figure supplement 1.** Comparison of mechano-chemo interaction trends across individual biochemical stimuli.

By comparing these reversed nodes to the network topology, we found that IL1 exerted both stimulatory effects via activation of AP1 and suppressive effects via activation of BMP and activin bound inhibitor (BAMBI), with suppressive behavior dominating procollagen I expression at high IL1 levels (*Figure 3H*). While low-tension levels allowed for IL1-mediated activation of AP1 and downstream procollagen expression, high levels of tension alone were sufficient to saturate AP1 levels, eliminating the stimulatory effects of IL1 and causing high cytokine doses to inhibit expression. Our model predicts a high degree of interaction between tension and biochemical stimuli, suggesting that the effects of current and potential therapeutics for cardiac fibrosis may be substantially altered by local tension.

## Network perturbation analysis reveals differential roles of tension

While targeted experimental studies of fibroblast signaling have identified numerous pathways mediating extracellular matrix turnover, it is still unclear whether certain mechanisms influence cell behavior more than others and whether these influences are altered with changes in the mechanical context. Using our systems-level model, we calculated changes in network activity following comprehensive knockdowns of individual nodes to functionally assess network-wide effects of perturbations. We calculated each node's 'knockdown influence' as the summed magnitudes of all changes in other nodes' activities upon knockdown of that particular node, and we calculated each node's 'knockdown sensitivity' as the summed magnitudes of all changes in that particular node's activity with knockdown of all other nodes (see Materials and methods, Network perturbation analysis section for full description). We repeated simulations for low, medium, and high levels of tension (reaction weights 0.25, 0.5, and 0.75, respectively) and selected for the top 10 scoring nodes in both influence and sensitivity, which indicated differentially responsive sets of nodes (*Figure 4A–C*, see *Figure 4—figure supplement 1* for network-wide sensitivity results).

At low levels of tension, knockdowns produced only small changes in node activities and correspondingly low influence and sensitivity values (*Figure 4A, D and E*). This is not surprising given

**Table 1.** Statistical tests of tension dependencies.
Results of Kolmogorov-Smirnov (K-S) tests with Benjamini-Hochberg correction comparing changes in area under normalized dose-response curves (ΔAUC) distributions for inputs across levels of tension.

| Input | Tension levels | K-S test p-value | Tension levels | K-S test p-value | Tension levels | K-S test p-value |
|---|---|---|---|---|---|---|
| AngII | 0.2 vs. 0.5 | 2.2e-17 | 0.2 vs. 0.9 | 2.37e-37 | 0.5 vs. 0.9 | 1.00e-32 |
| TGFβ | 0.2 vs. 0.5 | 1.88e-13 | 0.2 vs. 0.9 | 6.10e-32 | 0.5 vs. 0.9 | 5.20e-31 |
| IL6 | 0.2 vs. 0.5 | 1.87e-28 | 0.2 vs. 0.9 | 9.56e-6 | 0.5 vs. 0.9 | 9.90e-23 |
| IL1 | 0.2 vs. 0.5 | 5.38e-21 | 0.2 vs. 0.9 | 5.88e-16 | 0.5 vs. 0.9 | 5.88e-16 |
| TNFα | 0.2 vs. 0.5 | 3.17e-21 | 0.2 vs. 0.9 | 1.55e-39 | 0.5 vs. 0.9 | 1.55e-39 |
| NE | 0.2 vs. 0.5 | 1.07e-22 | 0.2 vs. 0.9 | 1.19e-3 | 0.5 vs. 0.9 | 7.47e-13 |
| PDGF | 0.2 vs. 0.5 | 5.38e-21 | 0.2 vs. 0.9 | 6.10e-32 | 0.5 vs. 0.9 | 6.10e-32 |
| ET1 | 0.2 vs. 0.5 | 1.15e-15 | 0.2 vs. 0.9 | 8.30e-34 | 0.5 vs. 0.9 | 1.70e-32 |
| NP | 0.2 vs. 0.5 | 1.18e-17 | 0.2 vs. 0.9 | 2.02e-12 | 0.5 vs. 0.9 | 6.97e-4 |

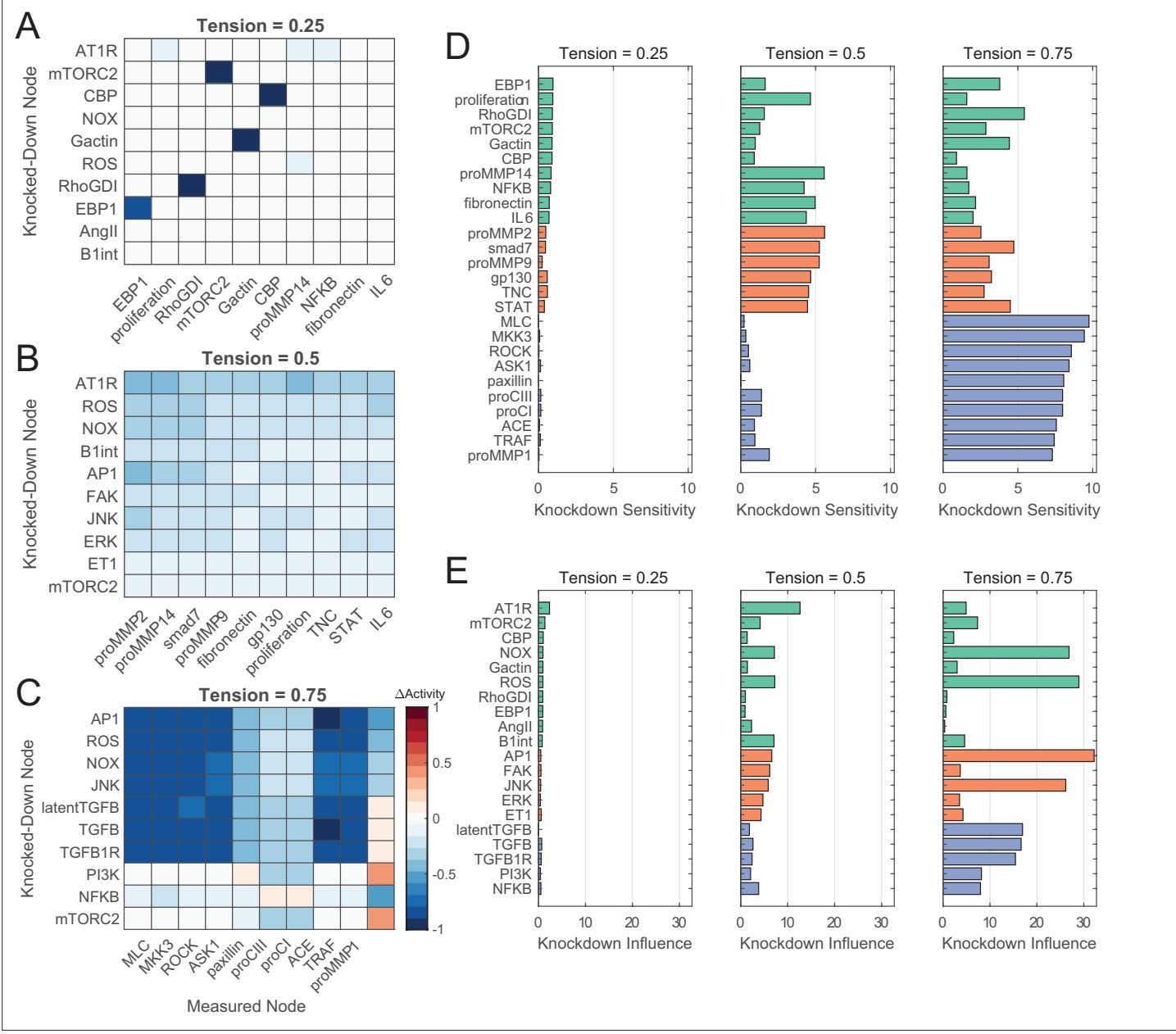

**Figure 4.** Network perturbation analysis reveals influential pathways of fibroblast mechanotransduction. (**A–C**) Changes in activity levels for top 10 sensitive nodes with simulated knockdown of top 10 influential nodes at low (**A**), medium (**B**), and high (**C**) levels of tension. Nodes were ranked based on knockdown influence (rows) and knockdown sensitivity metrics (columns), and changes in activity levels with perturbations are compared un-perturbed (i.e. no knockdown) conditions at each tension level. (**D–E**) Comparison of top-ranked sensitive and influential nodes between levels of tension. Nodes shown reflect all unique nodes that ranked in the top 10 based on knockdown sensitivity (**D**) or knockdown influence (**E**) between all levels of tension simulated, and colors reflect nodes that were top-ranked starting at low tension (green), medium tension (orange), and high tension (purple).

The online version of this article includes the following figure supplement(s) for figure 4:

**Figure supplement 1.** Full network perturbation analysis results.

all node activities are relatively low at this close-to-baseline context. In contrast, medium and high levels of tension demonstrated marked activity changes with knockdown perturbations and exhibited common influencers and unique sets of sensitive nodes. Under medium tension levels, knockdown of AT1R exhibited relatively broad inhibition compared to other top influencers, decreasing activity levels of several proMMPs, fibronectin, TNC, and members of the IL6 pathway by an average of 35.9% ±

5.4% (*Figure 4B*). While other top influencers mediated smaller reductions in activity across the top 10 sensitive nodes, NADPH oxidase (NOX), ROS, AP1, and c-Jun N-terminal kinases (JNK) demonstrated an increased capacity to mediate proMMP2, proMMP14, and smad7 levels under medium tension. Under high-tension conditions, however, these four nodes ranked as the highest influencers network-wide, mediating decreases in activity for members of the MAPK pathways (80.6% ± 4.3% reduction in activity on average), actin cytoskeleton-related signaling (68.7% ± 22.2% average reduction), and outputs procollagen I, procollagen III, and proMMP1 (30.9% ± 8.9% average reduction) (*Figure 4C*). Several members of the TGFβ1 pathway demonstrated similar trends in regulation including latent TGFβ1, suggesting that autocrine feedback may play a role in regulation at high tension.

Rankings of knockdown influence and knockdown sensitivity across all tension conditions suggest that influential mechanisms and their effects were dependent on the tensional context (*Figure 4D–E*). While ROS, AP1, JNK, and TGFβ1-related node knockdowns displayed an overall lower degree of influence on the network compared to knockdown of AT1R for low- and medium-tension conditions, knockdown of these nodes under high levels of tension mediated large changes in network-wide activity with knockdown influence levels increasing by up to 4.9-fold compared to medium tension (*Figure 4D*). Similarly, different sets of nodes were more sensitive toward perturbations between tension levels, with lower knockdown sensitivity levels of proMMPs 2, 9, and 14 and higher levels of proCI, proCIII, and proMMP1 at high tension relative to medium tension (*Figure 4E*). These behaviors suggest that tension acts via different pathways at different levels of stimulation, and cell behavior in a dynamic mechanical environment may not be dictated by a single mechanism alone.

## Fibroblast signaling network predicts behavior of current post-MI therapeutics

Current post-MI therapeutics aim to reduce cardiac hypertrophy via inhibition or activation of neuro-hormonal signaling (*Ponikowski et al., 2016*), both of which also alter cardiac fibroblast behavior and extracellular matrix turnover as a side effect. To test the hypothesis that our model can reproduce experimentally observed effects of current post-MI drugs on fibroblast behavior, we simulated the effects of three drug classes by modulating the maximum activation parameters of individual nodes: (1) angiotensin receptor blockers (ARB) were simulated via knockdown of the AT1R node, (2) neprilysin inhibitors (NEPi), which increase the bioavailability of NPs, were simulated via overexpression of the NP node, and (3) ARB-NEPi combination drugs were simulated via combination of (1) and (2) above (see Materials and methods, Drug effect comparisons section for full description and *Table 2* for simulation parameters).

We tested whether our model could discern both individual and synergistic effects of post-MI drugs by simulating dose-response relationships between the three drugs above and procollagen I expression. Von Lueder and colleagues previously found that valsartan primarily attenuated collagen expression in cardiac fibroblasts following exposure to AngII, with NEPi LBQ657 acting in a synergistic manner with valsartan but not altering collagen expression alone (*von Lueder et al., 2015*). Our model independently confirmed this conclusion, as increasing levels of AT1R knockdown combined with NP overexpression further attenuated procollagen I expression compared to AT1R knockdown alone (*Figure 5A*). We additionally observed that NP overexpression alone did not appreciably alter procollagen I expression while AT1R knockdown suppressed expression to near-control levels, further

**Table 2.** Model parameters used for drug effect simulations.

Minimum/maximum AT1R inhibitor (AT1Ri) values represent modifiers subtracted from the $Y_{max}$ parameter of the AT1R node, and minimum/maximum neprilysin inhibitor (NEPi) values represent modifiers added to the $Y_{max}$ parameter of the NP node. Non-specified input levels were set to a default value of 0.1. Int.: interpolated from measurements of post-MI cytokine levels in vivo.

| Study | TGFβ (w) | AngII (w) | NP (w) | Tension (w) | AT1Ri minimum | AT1Ri maximum | NEPi minimum | NEPi maximum |
|---|---|---|---|---|---|---|---|---|
| *von Lueder et al., 2015* | – | 0.4 | – | 0.3 | 0.01 | 0.5 | 0.05 | 4 |
| *Burke et al., 2019* | 0.4 | 0.4 | 0.6 | 0.3 | 0.5 | 0.5 | 4 | 4 |
| *Ramirez et al., 2014* | Int. | Int. | Int. | 0.1/0.6 | 0.5 | 0.5 | – | – |

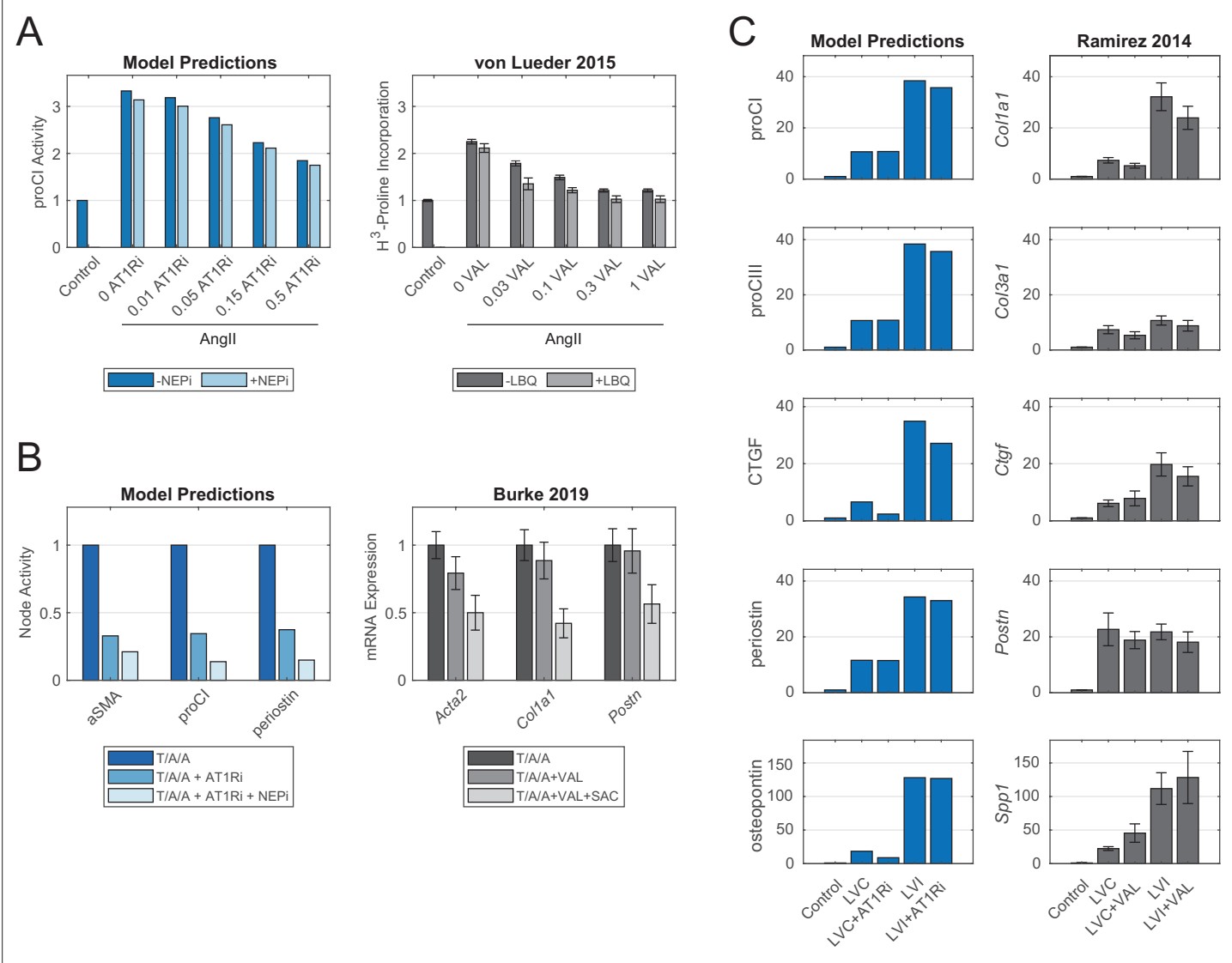

**Figure 5.** Model-predicted effects of current post-myocardial infarction (MI) drug treatments reflect experimental evidence from independent case studies. (**A**) Comparison of model-predicted effect of angiotensin receptor blocker (ARB) doses with or without neprilysin inhibitor (NEPi) treatment on procollagen I expression with measurements of collagen synthesis in cardiac fibroblasts in vitro. Doses for model predictions reflect negative modifiers of the $Y_{max}$ parameter of AT1R, doses for the in vitro study reflect doses of valsartan in μM, and values reflect fold changes compared to control conditions. (**B**) Comparison of model-predicted effects of ARB and ARB-NEPi treatments on three output nodes with corresponding gene expression measurements of cardiac fibroblasts in vitro. Values reflect fold changes compared to TGFβ1/AngII/atrial NP (T/A/A) conditions. (**C**) Comparison of model-predicted effects of ARB treatment on five output nodes in representative remote (LVC) and infarct (LVI) zones with corresponding gene expression measurements in post-MI mouse hearts. Values reflect fold changes compared to control (non-infarcted) conditions. All experimental data are represented as mean ± SEM.

The online version of this article includes the following figure supplement(s) for figure 5:

**Figure supplement 1.** Comparisons of model-predicted and experimentally measured changes in procollagen expression and signaling intermediate activity.

supporting the authors' conclusions of the dominant mechanism for combinatory ARB-NEPi drugs (*Figure 5—figure supplement 1*). This inability of NEPi class treatments to independently regulate AngII signaling can be attributed to a lack of overlap between AngII and NP pathways; AngII stimulation causes the activation of canonical and non-canonical TGFβ1 signaling via production of latent TGFβ1, and although NP overexpression inhibited canonical activation of smad3 by TGFβ1R, non-canonical activation and downstream procollagen expression via Akt remained unchanged.

Knockdown of AT1R inhibited both pathways of TGFβ1 signaling, thereby reducing procollagen I levels in a dose-dependent manner.

We next assessed our model's capability to predict the expression of multiple gene products and potential mechanisms-of-action following perturbation by the drug classes above. We compared model predictions of three model outputs following ARB or ARB-NEPi treatments to a recent study by Burke and colleagues, who observed synergistic inhibition of genes encoding alpha-smooth muscle actin (αSMA), collagen I, and periostin by combinations of valsartan and the NEPi sacubitril (*Burke et al., 2019*). We found that model simulations predicted a similar synergistic inhibition of all three gene products in high TGFβ1/AngII/atrial NP (T/A/A) conditions, and although simulations of ARB treatment alone reduced output expression as concluded in the von Lueder study, combinatory treatment in this context further reduced all three model outputs to an average 16.6% of positive controls (*Figure 5B*). The authors additionally suggested that sacubitril mediated on fibroblast behavior via activation of PKG and inhibition of the small GTPase Rho, with patient-derived fibroblasts responding to sacubitril or valsartan-sacubitril treatment with decreased fractions of active Rho (*Burke et al., 2019*). Upon incorporating this interaction into the model reaction logic, model-predicted activation levels of PKG and Rho following NEPi and ARB-NEPi treatments confirm this mechanism-of-action by NP-related treatments, with ARB-only treatments failing to change activity levels of both PKG and Rho compared to TGFβ1/AngII/NP controls (*Figure 5—figure supplement 1*). These comparisons demonstrate the ability of the model topology to predict both output-level changes with drug treatments as well as identify mechanisms-of-action compared to experimental data.

By incorporating known mechanotransduction pathways into network topology, we expanded our model's capability to predict responses to drug treatments in local cardiac tissue based on differences in tension. This capability is particularly important in the context of an infarcted left ventricle wherein fibroblasts in the infarct scar are subjected to heightened tensile stretches compared to the fibroblasts in the remote, non-infarcted myocardium that are subjected to normal myocardial tensions (*Torres et al., 2018*). We investigated this capability by simulating the expression of model outputs in response to valsartan treatment in both low- and high-tension contexts, while setting the model cytokine inputs to experimentally matched post-infarct levels as previously described (*Zeigler et al., 2020*). We compared model output expressions to a study by Ramirez and colleagues, who observed significant differences in fibrosis-related gene expression between infarct and remote zones and negligible effects of valsartan treatment alone using a mouse model (*Ramirez et al., 2014*). Model predictions generally agreed with their findings, as the degree of tension sizably altered the expression of procollagens, matricellular proteins, and connective tissue growth factor relative to changes in expression with valsartan treatment (*Figure 5C*). While we observed variable degrees of error between simulated levels of individual outputs and measured gene expression, our model accurately predicted qualitative differences in drug response based on regional tension.

## Computational drug screen predicts candidates for mechano-adaptive therapy

The infarcted ventricle presents a spatially dependent need to maintain extracellular matrix content at the infarct region while limiting scar formation in remote regions. While targeted experimental studies involving individual or combinatory post-MI drugs demonstrate efficacy either for individual cell types or bulk tissues, there is currently no evidence of current or potential drugs showing variable effects based on regional mechanics. Our analysis of mechano-chemo interactions above suggests that regional mechanics can alter the effects of positive and negative perturbations, so we investigated whether simulated drug treatments for low- and high-tension conditions could identify drug targets that provide regional therapeutic effects post-MI. Specifically, we sought potential drug targets that adapt matrix-related expression to the regional tension by increasing pro-matrix expression in high tension and decreasing expression in low tension. To that end, we performed a comprehensive screen of individual and combinatory drug targets involving single node knockdown and overexpression and dual node knockdown, overexpression, and knockdown-overexpression combinations for a total of 23,762 drug treatments.

We used a rank-based scoring strategy to score drug treatments based on changes in output expression compared to non-perturbed conditions, calculating a matrix content metric from individual output nodes and ranking treatments by the capacity to decrease or increase this metric with

low- or high-tension stimuli, respectively (see Materials and methods, Mechano-adaptive drug screen section for full description). While no individual drug target perturbation achieved these opposing responses (*Figure 6—figure supplement 1*), we found that 450 of 23,544 combinatory perturbations qualitatively altered matrix content based on regional tension, and we identified a cluster of 13 perturbations that scored highly in adapting expression of fibrosis-related outputs to regional tension (*Figure 6A*, boxed combinations). This cluster primarily involved dual knockdown of nodes related to the IL6 and Akt/mTOR signaling pathways, with 8 of 13 treatments targeting members of these pathways (*Figure 6B–C*). While most of these combinations exerted relatively minor changes in individual output levels compared to un-perturbed conditions, these combinations primarily acted to increase expression levels of procollagens, periostin, and PAI1 while decreasing proMMPs 1, 3, and 8 with high tension, thus aiding scar formation in high-tension conditions only (*Figure 6C*). Additionally, seven treatments showed evidence of increasing proMMP levels and decreasing procollagen and periostin levels with low tension (*Figure 6B*), suggesting that inhibition of the IL6 and Akt/mTOR pathways may simultaneously reduce pro-fibrotic matrix expression in remote tissues with low levels of tension. In follow-up in vitro experiments, we tested a subset of six combination perturbations by treating primary human cardiac fibroblasts cultured with pharmacological inhibitors under low mechanical stimulation (8 kPa stiffness plates) or high mechanical stimulation (64 kPa stiffness plates). Excitingly, four out of the six perturbations showed statistically significant differences in the drug treatment effects on collagen content (assessed as fluorescence intensity) with decreased collagen under low mechanical stimulation and increased collagen content under high mechanical stimulation (*Figure 6D*).

## Discussion

Spatial control of myocardial fibrosis after MI has remained a challenge due in part to simultaneous signaling from biochemical cues and regional mechanics, both of which mediate fibroblast activation and synthesis of extracellular matrix components (*Davis and Molkentin, 2014*). In the current study, we employed a computational model of fibroblast signaling to predict the regional expression of fibrosis-related proteins by incorporating known signaling pathways related to canonical biochemical agonists and mechanical tension. We found that the model correctly predicted qualitative changes in fibrotic protein expression with a similar degree of accuracy compared to other logic-based ODE networks (*Cao et al., 2020*; *Wang et al., 2020*), and model simulations of current therapeutics such as ARB and ARB-NEPi class drugs accurately reproduced trends in output gene expression observed in cardiac fibroblasts and a post-MI mouse model. Functional analyses identified influential mechano-chemo interactions and preferential pathways of mechanotransduction such as the production of latent TGFβ1 via a NOX-ROS-AP1 signaling axis. We additionally used the model as a targeted screening tool by simulating fibroblast responses to drug combinations in regions of low and high tension, and we identified 13 combinations that adapted the expression of matrix-related proteins to the regional mechanical environment. Our results suggest that this computational approach can aid the development of effective therapies for pathological tissue remodeling which involves simultaneous changes to paracrine signaling and tissue mechanics, and model predictions can generate targeted hypotheses for controlling fibroblast behavior for further experimental investigation.

### Crosstalk between biochemical and biomechanical factors in fibroblast signaling

Increased circumferential stretch and stiffness in the post-MI infarct zone have been established as both a direct regulator of fibroblast activation and an indirect regulator of biochemical signaling pathways via crosstalk mechanisms (*Herum et al., 2017b*; *Margadant and Sonnenberg, 2010*). Experimentally assessing the system-wide extent and dynamics of these interactions would be exceedingly challenging due to time and resource constraints; however, biological network models can overcome this challenge by comprehensively simulating biochemical dose-response relationships under a variety of mechanical contexts. Our findings indicate that tension exerts a wide influence on biochemical signaling in fibroblasts by either amplifying responses to biochemical stimuli in the case of NE, dampening responses in the case of TGFβ1, and even reversing responses in the case of IL1.

 The three example cases above share a common connection in tension-mediated expression and activation of latent TGFβ1, an autocrine feedback mechanism shown to influence cardiac fibroblast

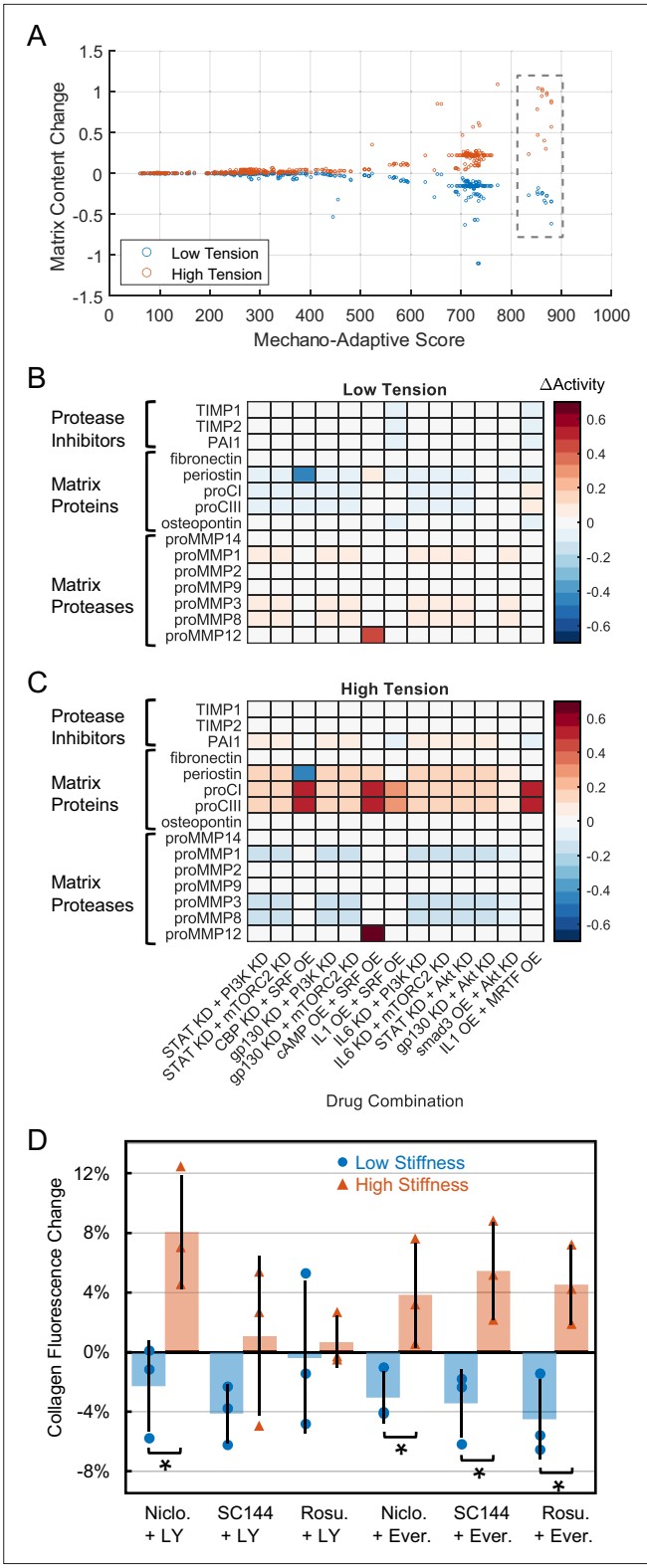

**Figure 6.** Comprehensive drug screen identifies mechano-adaptive candidates for post-myocardial infarction (MI) fibrosis. (**A**) Target combinations meeting preliminary threshold for mechano-adaptive behavior (i.e. promoted negative change in matrix content in low tension and positive change in matrix content in high tension) were scored based on rankings of matrix content changes in low- and high-tension simulations. Boxed values represent scores of 13 candidates examined in panels (**B**) and (**C**). (**B–C**) Changes in output expression predicted with

*Figure 6 continued on next page*

*Figure 6 continued*

mechano-adaptive candidates under tensional contexts, expressed as changes in node activity compared to non-perturbed conditions for each level of tension. Bracketed groups in panels (B/C) represent categories used for calculations of matrix content in (**A**). (**D**) Follow-up in vitro experiments treated human cardiac fibroblasts with pharmacological compound combinations that inhibited the model-identified targets. Four of six combinations successfully decreased collagen content under low mechanical stimulation (8 kPa stiffness) and increased collagen under high mechanical stimulation (64 kPa stiffness). * indicates $p<0.05$ using Student's t-tests with Bonferroni multiple comparison adjustments.

The online version of this article includes the following source data and figure supplement(s) for figure 6:

**Source data 1.** In vitro validation experimental results.

**Figure supplement 1.** Screen of individual drug targets for mechano-adaptive matrix expression.

phenotypes (*Hinz, 2015*; *Lindahl et al., 2002*). We found that NE acted to antagonize latent TGFβ1 expression via a cAMP-PKA-CREB signaling axis. Activation of CREB through the use of phosphodiesterase inhibitors has previously been shown to antagonize fibrotic expression of lung and cardiac fibroblasts (*Miller et al., 2011*; *Wójcik-Pszczoła et al., 2020*) as well as cardiac fibrosis in a pressure overload mouse model (*Gong et al., 2014*), suggesting a potential mechano-responsiveness of this mechanism. Simulated fibroblast responses to TGFβ1 indicated a synergistic response under tension with latent TGFβ1 activation and output expression occurring for lower input doses. This trend is consistent with experiments involving valvular myofibroblasts, which exhibited earlier and greater induction of bioactive TGFβ1 and aSMA protein levels when subjected to combined tension/TGFβ1 compared to either stimulus alone (*Merryman et al., 2007*).

We additionally found that stimulation with IL1 produced opposite effects depending on the level of tension, with IL1 increasing procollagen I expression for low- to medium-tension levels and decreasing expression for high-tension levels through activation of BAMBI and downregulation of downstream targets of TGFβ1R. As a positive regulator, IL1 upregulates MAPK and NFkB signaling pathways via myeloid differentiation factor-88 (MyD88) (*Van Tassell et al., 2015*), and both IL1R[-/-] and MyD88[-/-] mice display reduced fibrosis and progression toward heart failure in a model of acute myocarditis (*Blyszczuk et al., 2009*). As a negative regulator, IL1 upregulates the expression of BAMBI to act as a decoy receptor for TGFβ1, thereby reducing downstream signaling (*Saxena et al., 2013*). In BAMBI[-/-] mice, pressure overload worsened fibrosis compared to wild-type mice in a TGFβ1-dependent manner, suggesting that this negative regulation is dependent on both mechanical stress and TGFβ1 stimulation (*Villar et al., 2013*). IL1 has additionally been shown to reverse the myofibroblast phenotype in valvular interstitial cells with culture on stiff substrates (*Aguado et al., 2019*), further suggesting that context-dependent regulation by IL1 may be a determining factor of fibroblast phenotype. Our analysis of mechano-chemo interactions above suggests that computational approaches can reveal context-specific mechanisms of disease progression and reinforces the need to account for various environmental factors (such as local biomechanics) in developing therapeutic strategies.

## Influential mechanisms controlling fibroblast mechanotransduction

Fibroblasts transduce changes in their local mechanical environment using both well-known mechanosensors such as integrins and newly discovered mechanisms such as piezo channels and actin-dependent translocation of YAP/TAZ (*Herum et al., 2017b*). While these mechanisms have all been shown to alter cell phenotypes individually, it is still unclear as to whether individual pathways influence cell behavior over others. Network perturbation analysis predicted that tension is primarily mediated by an autocrine feedback mechanism in which a NOX/ROS/JNK/AP1 signaling axis stimulates latent TGFβ1 expression to drive further NOX activation as well as procollagen expression via smad3. The secondary activation of TGFβ1 has been demonstrated for various stimuli in fibroblasts such as TNFα and AngII (*Gao et al., 2009*; *Voloshenyuk et al., 2011*) as well as for other cell types by influencing changes in phenotype related to extracellular matrix deposition and angiogenesis (*Hamzeh et al., 2015*; *Joki et al., 2000*; *Zheng et al., 2001*). Inhibition of NOX2 and NOX4 in AngII-infusion mouse models has additionally been shown to attenuate interstitial collagen deposition, further implicating NOX/ROS signaling as a central influencer of cardiac fibrosis (*Johar et al., 2006*; *Zhao et al., 2015*). Our model prediction that this pathway primarily influences fibrosis-related protein expression suggests that perturbation of this pathway may induce larger reductions in cardiac fibrosis compared

to others, and experimental studies investigating this behavior could provide a basis for developing targeted anti-fibrotic therapies.

## Fibroblast network model as a screening tool for targeted therapeutics

The heterogenous nature of MI creates a clear need for regionally targeted therapies for tissue remodeling, with the infarct zone requiring scar formation to provide local structural integrity and remote zones requiring scar downregulation to prevent adverse tissue stiffening. Loss of cardiomyocytes additionally presents different mechanical conditions between the infarct and remote zones (*Torres et al., 2018*), and our model predictions above and experimental evidence suggest that myofibroblast-mediated tissue remodeling is dependent on this mechanical context (*Watson et al., 2012*; *Waxman et al., 2012*). Previous trials of anti-fibrotic therapies have demonstrated the need to account for local increases in circumferential stretch in infarcted tissue. Studies inhibiting known mediators of excessive fibrosis such as TGFβ1, osteopontin, and TIMP3 have shown increased risk of left ventricular thinning and cardiac rupture, early complications associated with impaired scar formation at the infarct zone (*Hammoud et al., 2011*; *Ikeuchi et al., 2004*; *Trueblood et al., 2001*) – an indication that therapies designed to inhibit global fibrosis are not sufficient to promote healthy scar formation where it is needed. A further complication of post-infarct fibrotic control is the lack of evidence of regional differences in tissue remodeling for current and potential drugs. While our comparisons of model-predicted and experimentally tested ARB and NEPi treatments above demonstrated the inhibitory effects of these drugs in a single context, not enough evidence exists to determine whether these drugs produce therapeutic effects in both remote and infarcted zones.

We investigated the ability of our model to predict regional effects of drug treatments based on regional mechanical tension, and our computational screen of 23,762 drug treatments yielded 13 candidates that adapted expression of extracellular matrix-related proteins to the regional tension. These candidates implicated the IL6 and Akt/mTOR signaling pathways as well as the IL1/cAMP and MRTF/SRF pathways as mediators of mechano-adaptive behavior. Perturbations involving the IL6 and IL1 inflammatory pathways individually have yielded mixed results clinically; while several animal studies and clinical trials perturbing IL6 and IL1 signaling have concluded that knockdown of these pathways suppresses cardiac fibrosis and mediates gain of function due to MI or factors associated with MI (*Abbate et al., 2013*; *Hilfiker-Kleiner et al., 2010*; *Ma et al., 2012*), others have concluded that IL6 and IL1 knockdown exacerbate fibrosis post-MI or in response to MI-associated biochemical factors (*Dziemidowicz et al., 2019*; *Obana et al., 2010*; *Turner, 2014*). The results from these studies indicate that the influence of these pathways on tissue remodeling is highly context dependent with varying responses between treatment time courses and methods of antagonism, and our drug screen indicates that mechanical signaling additionally modifies cellular and tissue responses to these drugs. While we found no clinical evidence of tissue remodeling in response to the model-predicted drug combinations above, these results suggest that pairing of anti-inflammatory treatments with those for additional signaling pathways may provide an additional advantage in controlling regional cardiac fibrosis, and our predictions provide testable hypotheses for further investigation. Moreover, this model-based approach toward target discovery provides substantial value for potential drug development by focusing efforts on drugs or drug combinations with strong mechanistic evidence of therapeutic efficacy. Combinatory therapies involving three or more drugs could additionally be discovered efficiently using global search algorithms. Such algorithms have been used previously to prioritize effective treatments within other biological network models without constraining the number of drugs used (*Weiss and Nowak-Sliwinska, 2017*), and these methods could be employed to compare global knockdown/overexpression combinations against a similar summary metric used above to describe ECM turnover. While global search methods can optimize drug combinations within a systems-level network, the large number of simulations required for these methods will necessitate either substantial computing resources for full systematic exploration/optimization or reduction of the search space for efficient discovery. Finally, our model and approach can additionally be expanded to investigate fibrosis across multiple disease states and for individual patient conditions to stratify patients based on predicted responses to current drugs or to identify drug candidates for subpopulations of patients.

## Study limitations and future directions

One major limitation of the current study stems from the need for quantitative predictions of fibroblast signaling processes and matrix production. While the current model was able to make semi-quantitative predictions of cell behavior based on network topology and normalized reaction parameters, data-driven reaction parameters are necessary both to fully investigate mechanisms-of-action of fibrotic activity and to relate signaling processes to experimental measures of fibrosis. Phospho-proteomic datasets generated using methods such as reverse-phase array provide direct analogs to intracellular protein activation or inhibition as well as the scope of measurements required for systems-level modeling, and previous studies have demonstrated the utility of these datasets in inferring cell signaling networks and fitting literature-curated network parameters (*Alexopoulos et al., 2010*; *Osmanbeyoglu et al., 2017*; *Terfve and Saez-Rodriguez, 2012*). Future implementation of in vitro or clinical phospho-proteomic data would improve the accuracy of model predictions and reveal additional mechanistic insight into fibroblast behavior in the post-MI environment.

An additional limitation of the current model is that of scope as it relates to both cell signaling processes and additional levels of regulation. While the current model was developed based on direct experimental evidence of signaling interactions stemming from selected biochemical and mechanical stimuli, we must acknowledge that additional stimuli and signaling pathways play a role in cardiac fibroblast signaling, such as interferon-γ and epidermal growth factor receptor signaling pathways (*Levick and Goldspink, 2014*; *Liu et al., 2018*). While not meeting our requirements for inclusion (see Materials and methods, Fibroblast signaling model development section for criteria), we expect these additional signaling mechanisms to aid in model predictions across the diverse range of stimuli represented in vivo.

It is important to note that a variety of biomechanical factors can potentially affect fibroblast behavior in the post-infarct environment including altered strains, stresses, and tissue stiffness (*Clarke et al., 2016*; *Richardson et al., 2021*; *Richardson et al., 2015*). In addition, both the magnitude and directionality of applied loads can differentially mediate cell behavior and matrix turnover (*Herum et al., 2017a*; *Wang et al., 2004*). In our model, we use 'tension' as a term to imply any change in molecular-scale force across cell receptor proteins and channels, which past experiments have shown can be driven by extracellular deformation (e.g. substrate strains that accompany tissue stresses) and intracellular acto-myosin contractility (e.g. pulling against a stiffened substrate). Undoubtedly, future work could help improve model predictions by experimentally calibrating how different cardiac fibroblast mechano-receptors may be more or less sensitive to different forms of mechanical stimulation.

Lastly, evidence suggests additional dimensions of cardiac fibrosis regulation including transcriptional regulation (*Lacraz et al., 2017*), extracellular remodeling of collagen fibers (*Richardson et al., 2015*), paracrine signaling involving inflammatory cells, immune cells and myocytes (*Mouton et al., 2018*), and additional modes of cell signaling (e.g. microRNA, matrix degradation products) (*Lindsey et al., 2015*; *Thum et al., 2008*). Several groups have made progress in modeling regulation within these levels in the heart (*Chen et al., 2019*; *Liu et al., 2021*; *Rouillard and Holmes, 2012*; *Skelly et al., 2018*; *Tan et al., 2017*; *Zeigler et al., 2020*). While incorporating all factors into a computational framework is a non-trivial effort, a combined, multi-scale approach capturing hetero-cellular communication and extracellular mechanics across a chronic time course will ultimately be necessary to fully predict tissue remodeling given the range of variables involved in post-infarct wound healing.

Through our simulated studies of fibroblast signaling and fibrotic activity above, we demonstrated the utility of computational models of mechano-chemo signal transduction in identifying mechanisms-of-action governing cell behavior in complex environments. While in vitro and in vivo evidence validating these results are crucial for translating this mechanistic insight into clinical solutions for cardiac fibrosis, this model-driven approach can be advantageous in focusing experimental efforts toward promising candidates that limit scar formation to regions where it is beneficial to do so. Excitingly, our simulations identified 13 novel drug target pairs whose modulation resulted in desirable mechano-adaptive effects – namely, an increase in matrix accumulation within the infarct zone (where scar material is beneficial to maintain structural integrity) and a simultaneous decrease in matrix within the remote zone (where scar material is detrimental to myocardial mechanical and electrical function). Given the broad significance of the involved pathways in many different cell types, our computational predictions also suggest that similar mechano-chemo-signaling interactions are likely present across many cell types and tissues, potentially confounding the translation of in vitro findings to in

vivo applications where the mechanical context is strikingly different. Therefore, predictive models integrating both modes of signal transduction represent a critical step toward discovering targeted therapeutics across many disease states involving complex environmental signals. Future phospho-proteomic studies of fibroblast signaling in combination with targeted studies of influential signaling pathways will both enable further development of targeted treatments for cardiac remodeling and enable therapeutic development for a range of fibrosis-related diseases.

## Materials and methods

### Data and code availability

The code generated during this study is available at GitHub (copy archived at swh:1:rev:e73f424b-d4477a1776d4316e3c5af8cd72c3b666; *Rogers, 2022*). The published article includes all model and validation databases generated or analyzed during this study.

### Fibroblast signaling model development

Our previously published model of cardiac fibroblast signaling was expanded using a manual literature search of over 100 peer-reviewed studies to include newly identified proteins and/or signaling pathways associated with fibroblast mechanotransduction, as well as several secreted proteins newly associated with cardiac fibrosis. New individual proteins (model nodes) and signaling reactions (edges) were identified from studies that concluded direct interactions between proteins altering protein activity (e.g. by phosphorylation). New nodes and/or edges were included if at least two independent studies contained evidence of either activation of a node or activation of another species by a node in either cardiac or other fibroblast subtypes. As previously described (*Tan et al., 2017*; *Zeigler et al., 2016*), the final network was implemented as logic-based ODEs in which activity levels of all nodes were modeled as a system of Hill equations, and all activity levels and input stimuli were represented as normalized values between 0 and 1. Logical NOT, AND, and OR gates were used for inhibitory and complex signaling interactions by applying logical operations $1 - f(x)$ , $f(x) f(y)$ , and $f(x) + f(y) - f(x) f(y)$ to Hill equations, respectively. The system of differential equations was constructed using the open-source Netflux package for MATLAB (https://github.com/saucermanlab/Netflux; *Kraeutler et al., 2010*), and all subsequent analyses were conducted using MATLAB (Mathworks, Natwick, MA). Visualizations of the full network and sub-networks were created using Cytoscape software (*Shannon et al., 2003*).

### Model validation

Qualitative changes in select model outputs and signaling intermediates were compared between model predictions and independent experimental studies. A set of studies independent from model development (47 papers total) were selected based on direct measurement of either output secretion or activity of a signaling intermediate in response to a single input stimulus in fibroblasts. Outputs included in this set were either collagens, matrix proteases, protease inhibitors, matricellular proteins, autocrine signaling species, or αSMA as a marker of fibroblast activation. Intermediates included were limited to members of canonical signaling pathways (i.e. Akt, CREB, ERK, JNK, p38, PKC, smad3, ROS), and only intermediates with direct measurements were included in the final set. Model predictions were performed by simulating basal conditions (all input weights = 0.1) for 80 hr, followed by simulating single input stimuli (weight = 0.8) for an additional 240 hr in order to allow for the system to reach steady-state activity levels. Because studies used for validation conducted in vitro experiments on tissue culture plastic, a tension input weight of 0.4 (40% of maximum stimulation) was used for all validation simulations. Changes in intermediate and output node activity from basal to stimulated conditions were binned into either 'increase', 'decrease', or 'no change' categories using ±5% change in activity levels, which has been shown previously to distinguish qualitative changes in behavior for single-stimulus simulations (*Tan et al., 2017*; *Wang et al., 2020*). To fit default model parameters to best describe experimental data, parameter sweeps of Hill coefficient ($n$) and half-maximal effective concentration (EC50) were conducted to maximize the accuracy of model predictions, using the percentage of correct predictions as a metric. Maximum activation levels ($Y_{max}$) were set to a default value of 1, and time constants ($\tau$) were set to default values of either 0.1, 1, or 10 corresponding with species involved in intracellular signaling reactions, ligand-receptor reactions, or

transcription-translation events, respectively. Reaction weights ($w$) were set to default values of 1 with the exception of autocrine feedback-associated reactions, which were set to default values of 0.8 due to the lowered probability of these reactions resulting diffusion of output proteins or sequestering by the extracellular matrix (*Hinz, 2015*). Values for $n$ and EC50 found to maximize predictive accuracy were used with the other described parameters for all subsequent simulations. All model species, reactions, and assigned parameters are detailed in *Supplementary file 1*.

## Mechano-chemo interaction analysis

Dose-response behaviors of biochemical stimuli were simulated for incremental levels of the 'tension' input reaction to assess interactions between mechanical and biochemical stimulations. For 0.1 increments in tension reaction weight ranging from 0.1 (10% of maximum stimulation) to 0.9 (90% of maximum stimulation), steady-state activity levels of all nodes in the network were simulated following stimulation of each biochemical input reaction (AngII, TGFβ1, IL6, IL1, TNFα, NE, PDGF, ET1, and NP) for 240 hr, ranging in reaction weight from 0 to 1 in 0.01 increments. The resulting dose-response curves were normalized to steady-state activity levels for a biochemical input of 0, and the total response of each node to a biochemical stimulus was calculated as the area under the normalized curve (AUC). Changes in AUC due to tension (ΔAUC) were calculated for each tension level as the difference between AUCs with added tension and with baseline tension (tension = 0.1). Sizeable mechano-chemo interactions were identified using a threshold of ±5% in ΔAUC. ΔAUCs greater than or equal to 5% were categorized as amplifying interactions (tension increases node response to a biochemical stimulus), ΔAUCs less than or equal to –5% were categorized as dampening interactions (tension decreases node response), and AUCs with opposite signs between baseline and tension conditions and exceeded the threshold of ±5% were classified as reversal interactions (tension causes biochemical stimulation to exert the opposite effect on node activity). This threshold has been shown previously to distinguish qualitative changes in activity levels with individual node perturbations (*Tan et al., 2017*; *Wang et al., 2020*), and while these studies did not use an integrated AUC metric for comparison, our comparison of AUCs requires greater peak activation levels to meet the threshold, thereby imposing a stricter comparison than these previous studies.

To compare the extent of interactions between tension levels for each input, two-sample Kolmogorov-Smirnov tests were conducted for distributions of ΔAUCs, comparing (1) distributions for tension = 0.2 vs. tension = 0.5; (2) distributions for tension = 0.2 vs. tension = 0.9; and (3) distributions for tension = 0.5 vs. tension = 0.9. Corrections for multiple comparisons were made using the Benjamini-Hochberg procedure (*Benjamini and Hochberg, 2009*), and adjusted p-values are displayed in *Figure 3—figure supplement 1*. As an additional measure supporting the input-related differences in interactions found above, topological network parameters were computed using the NetworkAnalyzer plugin for Cytoscape (*Doncheva et al., 2012*).

## Network perturbation analysis

To identify influential signaling mechanisms across various mechanical contexts, a series of node knockdowns were simulated for several levels of tension. For tension reaction weights of 0.25, 0.5, and 0.75, basal conditions were simulated for 80 hr (all other input weights = 0.1) followed by knockdown of individual nodes ($Y_{max} = 0.1$) for 240 hr. Steady-state activity levels of all nodes were measured across knockdowns of all nodes, and changes in activity (Δ*Activity*) were calculated as the difference between node activity after knockdown and basal activity. Knockdown sensitivity of each node was calculated as the total change in activity of the node across all knockdown simulations, and knockdown influence of each node was calculated as the total change in activity of all other nodes in the network upon knockdown.

## Drug effect comparisons

Predicted responses of the network to currently prescribed pharmacological therapies were compared three studies measuring cardiac fibroblast or myocardial tissue gene expression or protein expression in vitro or in vivo (*Burke et al., 2019*; *Ramirez et al., 2014*; *von Lueder et al., 2015*). All drugs were simulated by knockdown or overexpression of individual nodes: ARB valsartan (VAL), simulated by applying a negative modifier to $Y_{max}$ for the AT1R node, NEPi LBQ657 (LBQ) and sacubitril (SAC), simulated by applying a positive modifier $Y_{max}$ for the NP node, and angiotensin receptor NEPi LCZ696

(VAL+ LBQ) and sacrbitril/valsartan (VAL+ SAC). All negative control simulations were conducted using input reaction weights of 0.1, and all input reaction weights and drug perturbation levels for in vitro studies were chosen to maximize the dynamic range of network responses. Input reaction weights for in vivo studies were interpolated from experimental data of cytokine levels post-MI as previously described (*Zeigler et al., 2020*) using a time point of 4 weeks post-infarction, and tension reaction weights for control and infarct zones were set to 0.1 and 0.6, respectively. All input reaction weights and perturbation conditions can be found in *Table 2*.

## Mechano-adaptive drug screen

A computational screen of single and double network perturbations was conducted to identify drug targets with potential to adapt fibrotic activity to the local mechanical environment. Biochemical input reaction weights used for all simulations were interpolated using the method described for in vivo study comparisons at a time point of 2 weeks post-infarction. To represent remote (low-tension) and infarcted (high-tension) zones post-MI, tension reaction weights of 0.1 and 0.6 were used, respectively. For each simulation, baseline conditions were simulated using the input reaction weights above for 80 hr, and single or double perturbations were simulated by setting $Y_{max}$ values of nodes to 0.1 (knockdown) or 5 (overexpression) for 240 hr. Single perturbations were simulated for all individual nodes as both knockdown and overexpression for a total of 218 perturbations, and double perturbations were simulated for all node combinations as double knockdown, double overexpression, or knockdown-overexpression combinations for a total of 23,544 perturbations.

Perturbations that exhibited mechano-adaptive behavior were identified using a rank-based score metric based on changes in activity levels for model outputs (ΔActivity). For each perturbation and tension level, ΔActivity levels were calculated as the difference in each node's activity between baseline and perturbed conditions, and changes in overall matrix content (MCC) for each perturbation and tension level were calculated based on ΔActivity levels for all procollagens and matricellular proteins ($\Delta Activity_{Matrix}$), ΔActivity levels for all MMPs ($\Delta Activity_{MMP}$), and ΔActivity levels for all protease inhibitors ($\Delta Activity_{Inhib}$): $MCC = \sum \Delta Activity_{Matrix} - \Delta Activity_{MMP} + \Delta Activity_{Inhib}$. Output nodes were categorized as follows: $\Delta Activity_{Matrix}$: proCI, proCIII, fibronectin, periostin, osteopontin; $\Delta Activity_{MMP}$: proMMPs 1, 2, 3, 8, 9, 12, 14; $\Delta Activity_{Inhib}$: TIMP1, TIMP2, PAI1. To identify perturbations that (1) qualitatively alter matrix content in a desirable manner based on the tensional context and (2) maximize differences in low- and high-tension expression, a two-step procedure was used to score mechano-adaptive behavior from MCC values:

1. Perturbations were first filtered based on the numeric signs of MCC values at each tension level to identify categorically desirable behavior. Perturbations that exhibited negative MCC values (i.e. greater anti-fibrotic activity than pro-fibrotic activity) at low tension and positive MCC values at high tension (i.e. greater pro-fibrotic activity than anti-fibrotic activity) were retained for subsequent steps, and all others were removed.
2. Retained perturbations were then ranked based on MCC values at each tension level. MCC values at low tension were ranked in ascending order (i.e. low to high values), and MCC values at high tension were ranked in descending order (i.e. high to low). The mechano-adaptive score for each perturbation was then determined by summing each perturbation's rank for low and high tension and subtracting two times the sum from the number of retained perturbations, thereby

**Table 3.** Experimental group culture media conditions.

| Control | Media (0.5% FBS)+ TGFβ (10 ng/mL)+ TNFα (10 ng/mL)+ L-ascorbic acid (50 ng/mL) |
|---|---|
| PI3K + STAT inhibitors | Control + LY294002 (10 μM)+ Niclosamide (0.5 μM) |
| PI3K + gp130 inhibitors | Control + LY294002 (10 μM)+ SC144 (10 μM) |
| PI3K + IL6 inhibitors | Control + LY294002 (10 μM)+ Rosuvastatin (10 μM) |
| mTOR + STAT inhibitors | Control + Everolimus (0.5 μM)+ Niclosamide (0.5 μM) |
| mTOR + gp130 inhibitors | Control + Everolimus (0.5 μM)+ SC144 (10 μM) |
| mTOR+ IL6 inhibitors | Control + Everolimus (0.5 μM)+ Rosuvastatin (10 μM) |

denoting highly adaptive perturbations with high scores and less adaptive perturbations with low scores with a minimum score of 0.

## In vitro validation experiments for mechano-adaptive drug candidates

Primary human ventricular cardiac fibroblasts were purchased (Lonza #CC-2904) and cultured according to manufacturer's protocols with manufacturer-supplied growth medium (FGM-3 with 10% fetal bovine serum, fibroblast growth factor B, GA-1000). For testing drug effects under low mechanical stimulation vs. high mechanical stimulation, variable stiffness plates were purchased (8 kPa vs. 64 kPa CytoSoft Imaging 24-well plates, Advanced Biomatrix) and coated with collagen I according to manufacturer's protocols (PureCol, Advanced Biomatrix). Fibroblasts at passages 4–6 were seeded on collagen-coated plates at 25,000 cells per well, cultured in normal growth medium for 24 hr at 37°C and 5% $CO_2$, then treated with one of seven media conditions listed in *Table 3*.

After 72 hr of incubation with treatment media, cells were washed with PBS and stained for collagen I (Fisher #PA5-95137 primary antibody, Fisher #A-11012 Texas Red secondary antibody) and nuclei (DAPI), then imaged at 20× magnification with an EVOS FL-Auto inverted microscope. A total of 210 images were taken, accounting for seven media conditions × two stiffness conditions × three biological replicates/condition × five technical replicates/biological replicate. Images were background subtracted prior to calculating total collagen intensity values for each image, which were then averaged across technical replicates and normalized to the control condition for each stiffness plate (i.e. drug combinations on 8 kPa were normalized to 8 kPa control average, while drug combinations on 64 kPa were normalized to 64 kPa control average). Two-tailed Student's t-tests were made between the normalized effects of each drug on 8 kPa vs. 64 kPa plates with p-values adjusted for multiple comparisons using a Bonferroni correction, and $p < 0.05$ designated as significant.

## Acknowledgements

This study was supported by grants from the National Institutes of Health (GM121342, HL144927) and the American Heart Association (17SDG33410658). We would like to thank the Cyberinfrastructure Technology Integration group at Clemson University for generous allotment of compute time on the Palmetto high-performance computing resource. We would also like to thank Dr Jeff Holmes at the University of Alabama – Birmingham and Dr Jeff Saucerman at the University of Virginia for rich and thoughtful discussion around the ideas motivating this work.

## Additional information

### Funding

| Funder | Grant reference number | Author |
|---|---|---|
| National Institute of General Medical Sciences | GM121342 | William J Richardson |
| National Heart, Lung, and Blood Institute | HL144927 | William J Richardson |
| American Heart Association | 17SDG33410658 | William J Richardson |

The funders had no role in study design, data collection and interpretation, or the decision to submit the work for publication.

### Author contributions

Jesse D Rogers, Conceptualization, Data curation, Formal analysis, Investigation, Methodology, Resources, Software, Validation, Visualization, Writing - original draft, Writing – review and editing; William J Richardson, Conceptualization, Funding acquisition, Methodology, Project administration, Resources, Supervision, Visualization, Writing – review and editing

**Author ORCIDs**

William J Richardson  http://orcid.org/0000-0001-8678-9716

**Decision letter and Author response**

Decision letter https://doi.org/10.7554/eLife.62856.sa1
Author response https://doi.org/10.7554/eLife.62856.sa2

## Additional files

### Supplementary files

• Supplementary file 1. Fibroblast mechano-chemo signal transduction model. Database detailing all nodes, reactions, model parameters, and references used for construction of the signaling network.

• Supplementary file 2. Model validation database. Database detailing all qualitative input-output and input-intermediate relationships and references used for validation of the signaling network with independent experimental studies.

• Transparent reporting form

### Data availability

Our model, datasets used for analysis, and scripts necessary to reproduce all analysis and figures are freely available on GitHub (https://github.com/SysMechBioLab/Fibroblast_Signaling_Network_Model, copy archived at swh:1:rev:e73f424bd4477a1776d4316e3c5af8cd72c3b666).

The following dataset was generated:

| Author(s) | Year | Dataset title | Dataset URL | Database and Identifier |
|---|---|---|---|---|
| Richardson WJ | 2022 | Fibroblast_Signaling_Network_Model | https://github.com/SysMechBioLab/Fibroblast_Signaling_Network_Model | GitHub, e73f424 |

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
