## [Editor Report]

This paper presents a computational network model of fibroblast signalling in order to identify drug combinations that might be useful targets for controlling cardiac fibrosis, with application to treating myocardial infarction. The approach clearly has merit and is a potentially powerful way to make progress in understanding effects of interventions in situations where regional variations in tension and biochemical alterations have to date frustrated many attempts of understanding and rational treatment.

---

## [Decision Letter]

**Decision letter after peer review:**

Thank you for submitting your article "Fibroblast Mechanotransduction Network Predicts Targets for Mechano-Adaptive Infarct Therapies" for consideration by *eLife*. Your article has been reviewed by 3 peer reviewers, including Jennifer Flegg as Reviewing Editor and Reviewer #1, and the evaluation has been overseen by Aleksandra Walczak as the Senior Editor. The following individual involved in review of your submission has agreed to reveal their identity: Andrew McCulloch (Reviewer #2).

The reviewers have discussed the reviews with one another and the Reviewing Editor has drafted this decision to help you prepare a revised submission.

Summary:

The authors extended an existing systems model of cardiac fibroblast pro-fibrotic cell signaling network and used it to investigate interactions between the effects of altered tension and ligand-mediated signaling that could identify novel combination drug targets that could be used to target excessive collagen matrix accumulation in remote myocardium while not inhibiting ECM expression in the myocardial required for ventricular structural integrity. Model validation with independent experimental data showed 80% accuracy, which is very encouraging and comparable to previous reports with this class of cell systems model. The authors then looked at how sensitivity of network responses to node knockdown was affected by tension and used the model to identify subnetwork interactions mediating significant crosstalk. Model-predicted responses to combination treatment with angiotensin receptor blockers and neprilysin inhibition showed good agreement with published data. The authors concluding by predicting drug combinations that could be modulated significantly by tension.

Essential Revisions:

1. More motivation on why the methods applied here are appropriate over others.

2. The contribution over the previous work should be made clearer.

3. Is it possible to do a formal fitting or sensitivity analysis for the many model parameters?

4. Can the authors comment on whether therapies with 3 or more drugs can be combined and, if so, could the optimal treatment be found with optimisation methods?

5. The authors should be more specific about exactly whether they interpret tension as stress or strain based on the experiments used to formulate the model and they should identify experiments that enable the predictions to be tested less ambiguously. If these independent experiments cannot be found, then the paper would be improved by new in-vitro experiments that test some subset of the predicted tension-dependent drug combinations directly. The potentially important role of paracrine interactions should also be discussed.

6. Given the fairly strong emphasis of the manuscript on the identified 13 candidates that are predicted to have differential effect on pro-matrix expression depending on local tension, doesn't this require a minimum of a proof-of-concept experiment to illustrate both the types of experiments, say using cultured cells, that need to be performed and actually test if these predictions are realistic?

7. While the authors note a number of limitations, with the strong emphasis relating this study to cardiac infarcts, the omission of major cell types (such as a complete lack of cardiomyocytes) in the current model may leave this as too preliminary and incomplete to construct such a strong link with the infarct situation. For example, certain aspects of feedback between cell systems cannot currently be captured as only fibroblasts are modelled. It may require at least a toning down of the interpretation of the data shown here.

---

## [Author Response]

Essential revisions:1. More motivation on why the methods applied here are appropriate over others.

Clarifying the advantages of our methods is indeed an important point, so we have added a paragraph to our ‘Introduction’ section that summarizes the advantages of our modeling approach.

2. The contribution over the previous work should be made clearer.

As stated in paragraph 1 of ‘Results’, the advances over the previously published fibroblast signaling model focused on (1) adding a substantial number of mechanotransduction signaling nodes and (2) adding multiple protein outputs that have been shown to mediate cardiac fibrosis including proteases, matrix proteins, and matricellular proteins. To help clarify these additions, we included Figure 1—figure supplement 1, which highlights the specific node additions and reaction additions between our previous work (Zeigler 2016) and our current model. Apart from advances in the model topology, we performed all new simulations for this paper in order to explore specific interactions between mechanical stimulation and biochemical stimulation as detailed in Figures 3-6.

3. Is it possible to do a formal fitting or sensitivity analysis for the many model parameters?

As stated in paragraph 2 of ‘Results’, we did conduct parameter sweeps for half-maximal effective concentrations (EC50) and Hill coefficients (n) with results shown in Figure 2—figure supplement 1. We did not perform a parameter fitting process for all individual reaction parameters (e.g., each individual reaction’s EC50, n, and reaction weight) because this process would require more comprehensive and uniform datasets than what we have found in the literature. We did perform global sensitivity analysis in the sense of testing the influence of each molecular species in the model using a knockdown perturbation process, and those results are included in Figure 4—figure supplement 1.

4. Can the authors comment on whether therapies with 3 or more drugs can be combined and, if so, could the optimal treatment be found with optimisation methods?

Combinatory therapies involving 3 or more drugs could potentially be discovered using a similar approach to the mechano-adaptive drug screen described here. Global search methods such as genetic particle swarm algorithms could additionally be employed to compare global knockdown/overexpression combinations against a similar summary metric used here describing ECM composition, and such approaches have been used to discover drug combinations that are efficacious in other disease contexts without constraining the number of drugs used (Weiss and Nowak-Sliwinska, 2017). While such methods could prove useful for optimizing the desired therapeutic outcome, the increasing number of simulations will require either substantial computing resources for full exploration/optimization or reduction of the search space for efficient discovery. We have updated the Discussion section to include this commentary on future uses of network models to identify more complex drug combinations beyond our current study.

5. The authors should be more specific about exactly whether they interpret tension as stress or strain based on the experiments used to formulate the model and they should identify experiments that enable the predictions to be tested less ambiguously. If these independent experiments cannot be found, then the paper would be improved by new in-vitro experiments that test some subset of the predicted tension-dependent drug combinations directly. The potentially important role of paracrine interactions should also be discussed.

We apologize for the lack of clarity on this very important point. In general, we do not believe that cells respond directly to stress or strain but, rather, cells respond to molecular conformation changes that result in different molecular binding affinities and downstream biochemical signals. To that end, we intentionally used the word ‘tension’ as a term for implying changes in molecular-scale force across cell receptor proteins and channels. Past experiments have shown this tension can be driven by both extracellular deformation (via substrate stress and accompanying strain) and intracellular acto-myosin contractility (via pulling against a stiff substrate). The papers used to build our mechanotransudction network included some experimental results from stress/strain-induced stimulation and other experimental results from stiffness-induced stimulation. We feel this combination of diverse mechano-stimuli is appropriate since (in our opinion) the field has not comprehensively parsed out whether different cardiac fibroblast mechano-receptors are more or less sensitive to these different mechano-stimuli. Certainly, future work could help improve this understanding by teasing apart specific cell responses to varying stress/strain/stiffness in various combinations. To help clarify this point in the manuscript, we have added related text in the limitations at the end of our ‘Discussion’ section.

We completely agree that new in vitro experiments testing a subset of the model-predicted tension-dependent drug combinations would improve this paper. To that end, we have now included new experimental results in Figure 6 with accompanying text in the ‘Results’ section.

Regarding paracrine interactions, we are not sure what the reviewer specifically has in mind for this critique? Our fibroblast signaling model’s biochemical inputs (e.g., inflammatory cytokines) are directly capturing the paracrine signals being secreted by nearby cells (macrophages, neutrophils, cardiomyocytes) as well as autocrine signals being secreted by the fibroblasts themselves, which are captured in our model by biochemical feedback loops.

6. Given the fairly strong emphasis of the manuscript on the identified 13 candidates that are predicted to have differential effect on pro-matrix expression depending on local tension, doesn't this require a minimum of a proof-of-concept experiment to illustrate both the types of experiments, say using cultured cells, that need to be performed and actually test if these predictions are realistic?

We agree this was a shortcoming of the original submission and have now added new in vitro validation experiments for a subset of the 13 drug combinations, as shown in Figure 6.

7. While the authors note a number of limitations, with the strong emphasis relating this study to cardiac infarcts, the omission of major cell types (such as a complete lack of cardiomyocytes) in the current model may leave this as too preliminary and incomplete to construct such a strong link with the infarct situation. For example, certain aspects of feedback between cell systems cannot currently be captured as only fibroblasts are modelled. It may require at least a toning down of the interpretation of the data shown here.

This is an important point, so we have added this as a future direction at the end of our discussion. Still, given that most cardiomyocytes are killed and absent within a healing infarct (leaving predominantly fibroblasts in the long-term), we feel our presented model adds tremendous knowledge to the field’s understanding of fibrotic signaling in the infarcted ventricle. We also note that the in vivo experimental data used herein for validation (Figure 5) focused on matrix molecules which have been shown to be produced primarily by the fibroblasts rather than cardiomyocytes. Lastly, we note that this fibroblast model alone has previously proved sufficient for successfully capturing post-infarct inflammation dynamics (Ziegler et al., 2020). Ongoing work integrating our fibroblast model with published cardiomyocyte and macrophage models will be an important advance in the future (Tan et al., 2017, Ryall et al., 2012 and Liu et al., 2021).